# Structural Modifications and Novel Protein-Binding Sites in Pre-miR-675—Explaining Its Regulatory Mechanism in Carcinogenesis

**DOI:** 10.3390/ncrna9040045

**Published:** 2023-08-10

**Authors:** Abhishek Dey

**Affiliations:** Department of Biotechnology, National Institute of Pharmaceutical Education and Research (NIPER-R)-Raebareli, Lucknow 226002, India; rf.abhishek.dey@niperrbl.ac.in or 41.abhishek@gmail.com

**Keywords:** pre-miR-675, H19 lncRNA, miRNA, SHAPE assay, interactome, carcinogenesis, tumours

## Abstract

Pre-miR-675 is a microRNA expressed from the exon 1 of H19 long noncoding RNA, and the atypical expression of pre-miR-675 has been linked with several diseases and disorders including cancer. To execute its function inside the cell, pre-miR-675 is folded into a particular conformation, which aids in its interaction with several other biological molecules. However, the exact folding dynamics of pre-miR-675 and its protein-binding motifs are currently unknown. Moreover, how H19 lncRNA and pre-miR-675 crosstalk and modulate each other’s activities is also unclear. The detailed structural analysis of pre-miR-675 in this study determines its earlier unknown conformation and identifies novel protein-binding sites on pre-miR-675, thus making it an excellent therapeutic target against cancer. Co-folding analysis between H19 lncRNA and pre-miR-675 determine structural transformations in pre-miR-675, thus describing the earlier unknown mechanism of interaction between these two molecules. Comprehensively, this study details the conformation of pre-miR-675 and its protein-binding sites and explains its relationship with H19 lncRNA, which can be interpreted to understand the role of pre-miR-675 in the development and progression of tumorigenesis and designing new therapeutics against cancers.

## 1. Introduction

microRNAs (miRNAs) are small non-coding RNA which post-transcriptionally regulate gene expression by interacting with mRNA in multicellular organisms. Their interaction with mRNAs affects the stability and translation of the latter, thus modulating multiple processes including proliferation, differentiation, and apoptosis etc. Pre-miR-675 is a 73 nucleotide-long micro-RNA, which is expressed from the exon 1 of H19 long noncoding RNA (lncRNA). H19 lncRNA is temporally expressed in humans, where it is expressed during the placental stage, while the expression is drastically reduced after birth [1]. Interestingly, the abnormal expression of H19 lncRNA has been associated with various diseases in humans, including tumorigenesis, neurogenesis, and angiogenesis [1].

Being expressed from H19 lncRNA, the aberrant expression of pre-miR-675 has also been associated with various disorders [2]. Enhanced expression of pre-miR-675 is associated with the indefinite proliferation and growth of hepatic cells, resulting in hepatocellular carcinoma [3]. Studies have shown that miR-675-3p (a mature form of miR-675) can promote colorectal cancer and esophageal squamous cell carcinoma and regulate skeletal muscle regeneration and differentiation [4,5,6]. Recently, an association between abnormal expression of pre-miR-675 and cardiovascular disease has also been identified [7]. Elevated levels of pre-miR-675 in the patient serum suffering from atherosclerosis highlight the need to use pre-miR-675 as a new diagnostic biomarker in early diagnosis of cardiovascular disorders [7]. Abnormal expression of pre-miR-675 has been linked to various neurological diseases. The downregulated expression of pre-miR-675 has been found to be associated with the cases of treatment-resistant schizophrenia (TRS) [8]. H19 lncRNA, the precursor of pre-miR-675, was also found to be highly activated in human dopaminergic neuron loss, which results in Parkinson’s disease [9]. In contrast, downregulated H19 lncRNA, along with upregulated miR-129, was found to rescue PC12 cells, mimicking Alzheimer disease by increasing cell viability and restricting apoptosis [10].

miRNAs regulate gene expression either by repressing translation or by promoting mRNA degradation by binding to their 3′ UTR. Any unusual expression of miRNA regulates its association with different biological interactors, resulting in disease progression and pathology. Multiple studies have indicated the interplay between pre-miR-675 and interactors during the advancement of cancer and tumorigenesis. Higher plasma levels of pre-miR-675 and its precursor H19 lncRNA in the breast cancer patients indicate an interplay between both miRNA and lncRNA. Indeed, the overexpression of pre-miR-675 was able to mitigate the inhibitory effect of siRNA on H19 lncRNA in breast cancer cell line MCF-7 [11]. Interaction of pre-miR-675-5p with serine–threonine phosphatases have been shown to activate the Glycogen Synthase Kinase 3ꞵ (GSK 3ꞵ), thus enhancing the nuclear localization of ꞵ-catenin, which aids in the progression of colorectal cancer [12]. Direct interaction between pre-miR-675 and mRNA of ubiquitin 3 ligase family (c-Cbl and Cbl-b) have shown to enhance tumorigenesis and metastasis in breast cancer cells [13]. Contrary to the progression of tumours, pre-miR-675 has also been shown to suppress tumour growth by binding to Fas-associated protein with death domain (FADD) and inhibiting its activity in cellular apoptosis, thus promoting necroptosis in human hepatocellular carcinoma cell lines [14].

In cell, miRNAs also work in tandem with different RNA-binding proteins (RBPs) to regulate gene expression. Binding of RBP to miRNA enhances their biological activity either by participating in their biogenesis or facilitating their binding to 3′ UTR of target mRNA. The overexpression of DICER1 protein, an essential enzyme involved in the processing of miRNA, results in the progression of acute myeloid leukaemia (AML) [15]. Similarly, the repression of several miRNA biogenesis components due to alcohol and smoking causes the onset of hepatocellular carcinoma [16]. Alternatively, direct interactions between miRNA and RBPs also increase the function of miRNA by enabling them to access the less available targets. Pumilio RBPs like PUM1 and PUM2 interact with several miRNAs and increase their biological activity by enabling them to identify their respective target site, resulting in the development of various cancers [17,18,19]. Moreover, interplay between various miRNAs and lncRNA are known to regulate each other’s activities. miR-675 can epigenetically upregulate early growth response gene 1 (EGR1), which in turn increases H19 expression in liver cancer [20]. However, the detailed molecular mechanism behind the direct interactions between pre-miR-675 and RBP or its parent H19 lncRNA is currently unknown, specifically during cancer progression. Despite the advancement in determining the role of pre-miR-675 in the progression and metastasis of various tumours, its precise mechanistic function and regulatory effect in carcinogenesis is still obscure. Micro RNAs are known to fold into specific conformations, which are essential for their interactions with specific proteins/interactors and hence their function. Regardless of the in-depth biochemical analysis of pre-miR-675 and its involvement in various tumours, the knowledge about its detailed folding dynamics, interacting partners and the RNA-binding protein (RBP)-interacting sites on pre-miR-675 is still missing. Understanding how the folding of pre-miR-675 contributes its crosstalk with various proteins and RNA will be key in defining the involvement of pre-miR-675 in the development and progression of diseases like cancers. Here, we report the very first detailed conformational analysis of pre-miR-675 using a selective 2′ hydroxyl acylation analysed by primer extension (SHAPE) assay. Novel protein-binding hotspots on pre-miR-675 were identified using the RBPsuite algorithm. The co-fold analysis of pre-miR-675 with its precursor H19 lncRNA was also undertaken to understand any conformational changes incurred on both interactors upon their binding that could explain the regulatory role of pre-miR-675 in the progression of tumorigenesis. This current study reports the detailed structural analysis of pre-miR-675. Understanding its conformation and protein-binding sites and performing co-fold analysis with H19 lncRNA further underscores the therapeutic importance of pre-miR-675 in cancer biology.

## 2. Results

### 2.1. Secondary Structure of Pre-miR-675 Represents Canonical Stem Loop Helical Conformation

To experimentally analyse the folding conformation of nascent pre-miR-675, mutational profiling was performed after chemically modifying in vitro-transcribed pre-miR-675 with 5-Nitro Isatoic Anhydride (5NIA). Being a SHAPE reagent, 5NIA is known to modify the backbone of RNA molecules by forming a chemical adduct with the 2′ hydroxyl group of the unpaired RNA nucleotides [21]. These modifications can be identified through mutational profiling followed by massively parallel sequencing. SHAPE analysis of the two independent replicates (*n* = 2) of in vitro-transcribed pre-miR-675 resulted in chemical reactivity profile for both replicates with most of the nucleotides having low chemical modification, except for nucleotides G12, U13, A18, A34, C35, U36, U37, G38, G39, U40 and A49 (Figure 1A,B). This suggests that these nucleotides are highly flexible and are not involved in base-pairing interactions. Comparative analysis showed that both replicates have identical chemical reactivity profiles (Pearson’s correlation coefficient, R = 0.98) (Figure 1C,D), while the Scorer analysis [22] on the models obtained from replicates showed that both RNA folds into an identical conformation with the positive predictive value (PPV) and sensitivity (Sens) of 100% (Figure 1E). Positive predictive value (PPV) is the fraction/percentage of the base pairs in the predicted minimum free energy (MFE) structure where the known structure is determined through comparative sequence analysis. Sensitivity (Sens) is the maximum expected accuracy of RNA structure determined by a fraction/percentage of known base pairs correctly predicted in a minimum free energy (MFE) structure, where the known base pairs are determined by comparative sequence analysis. Once it was confirmed that both replicates’ folds into an identical conformation, average nucleotide chemical profiling was generated (by merging the fastq files for both replicates and re-running the shapemapper algorithm on the merged fastq files) and the final structure was created for pre-miR-675 (Figure 1F,G).

Based on the SHAPE constraints and minimum free energy, pre-miR-675 folds into a stem loop helical conformation, which includes multiple internal bulges or loops and culminates in an apical loop. The basal stem of the pre-miR-675 helix was stabilised by seven base pairs (nts 1–7 and nts 67–70), which includes six GC base pairs (Figure 1G). There are three internal loops (Loop I, Loop I, and Loop III) and a single bulge (Bulge I). Loop I and Loop II are formed by two unpaired or flexible nucleotides (U8 and C67—Loop I) and (A18 and A57—Loop II) (Figure 1G), while Loop III was the largest internal loop, which was formed by four unpaired nucleotides (C25, C26, A49 and U50) in the stem loop helix structure of pre-miR-675. A single nucleotide bulge (U13) was also present, which was sandwiched between internal Loop I and Loop II. Interestingly, only nucleotides U13 (Bulge I), A18 (Loop II) and A49 (Loop III) were found to be accessible to the SHAPE reagent, 5NIA, thus suggesting that the remaining inaccessible loop nucleotides were either arranged alternatively or involved in an intermolecular interaction, thus resulting in minor conformations that are energetically least favourable and hence remain undetected in the current study. The apical end of the pre-miR-675 ends with a formation of a 10-nucleotide apical loop starting from G33 to A42 (5′-GACUUGGUGA-3′) (Figure 1G). Most of the apical loop nucleotides were highly reactive to 5NIA, thus suggesting that they were not involved in any base pair interactions and were highly flexible. A hexanucleotide base pair forms a bridge between internal Loop 3 and terminal apical loop (Figure 1G). Overall, the chemical probing strategy of pre-miR-675 reveals a tight canonical stem–loop helical conformations with most nucleotides involved in base pairing interactions being inaccessible to 5NIA.

### 2.2. Pre-miR-675 Acts as a Hotspot for Many RNA-Binding Protein

microRNAs are known to regulate gene expression by interacting with various biomolecules inside the cells. Any abnormalities in this interaction will result in altered gene expression and disease conditions. Multiple studies have identified various interactors of pre-miR-675, however the overall interaction network of pre-miR-675 is still obscure. Within a cellular system, RNA-binding proteins (RBPs) are involved in multiple biological processes, including gene expression and the regulation of cellular pathways. Determining the RBP binding site on RNA is essential for gaining a mechanistic understanding of above processes, which are regulated through microRNAs or non-coding RNAs. RBPsuite was used to determine the pre-miR-675 binding proteins and the motif/sequences involved in RNA–protein interactions. Identified pre-miR-675 binding proteins were segregated based on binding scores (Appendix A) and only proteins with binding scores of more than 0.5 are reported here (Appendix A). Proteins with verified motifs for pre-miR-675 are also reported with sequence logos (Figure 2). The obtained sequence was further mapped on the stem loop helix model of pre-miR-675 (Figure 2). Interestingly, most of the motifs were found to be assembled from nucleotides, which either constitute bulges, internal loops, or the apical loop of pre-miR-675. FUS protein motif was found to be located on Bulge I of pre-miR-675 (Figure 2A,G). Surprisingly, for proteins FXR2, Lin28B and SRSF1, RBPsuite identified a common motif spanning from G16–G23 for FXR2 and Lin28B (Figure 2) and A18–C26 for SRSF1 (Figure 2). Nevertheless, this region also contains nucleotide A18, which is the part of Loop II in the stem loop helix model of pre-miR-675. The motif for SRSF9 was detected at the downstream end of SRSF1 motif spanning from A27–A35, which also includes a part of the apical loop (Figure 2). The apical loop of pre-miR-675 was also found to harbour the binding site for Human Antigen R (HuR) protein (Figure 2F,G). Overall, the search resulted in an array of pre-miR-675 interactors and their binding motifs in pre-miR-675, which can be further utilised to perform structure-based drug design.

### 2.3. H19 Long Noncoding RNA (lncRNA) Sequester Pre-miR-675 by Modifying Its Conformation

Although miRNAs regulate various cellular activities by interacting with a wide array of biological molecules, their activities are themselves controlled by the sponging effect of lncRNA, where lncRNAs can interact directly with miRNA and sequester them from their respective pathways. It has been known for a while that H19 lncRNA can interact with pre-miR-675 [11], however, how H19 lncRNA and pre-miR-675 crosstalk is currently unknown. To understand the nature of interactions between pre-miR-675 and H19 lncRNA, RNAcofold followed by comparative base pair conservation analysis between individual RNA and co-folded hybrid molecules was performed. Surprisingly, it was found that all the H19 exons interact with pre-miR-675 and these interactions rearrange the intramolecular base pair interactions within both H19 lncRNA and pre-miR-675 itself (Figure 3 and Appendix A). Interactions between H19-exon1 and pre-miR-675 resulted in two identical heterodimeric conformations with different binding energies (Appendix A). Closer examination of both models suggests that the 5′ end of pre-miR-675 interacts with the 3′ end of H19-exon1. This hybridization resulted in the global base pair rearrangements and opening of pre-miR-675, thus forming a chimeric RNA molecule conjugated of H19-exon1 and pre-miR-675 (Figure 3A, Figure 4A and Appendix A). Similarly, interactions between H19-exon3 and pre-miR-675 also resulted in two identical heterodimeric conformations with global rearrangements in nucleotide pairing (Figure 3C and Appendix A). This rearrangement also resulted in the opening of pre-miR-675 conformation (Figure 3C) and the binding of 5′ end of pre-miR-675 to 3′ end of H19-exon3 lncRNA, thus again forming a hybrid RNA conformer (Figure 4A). Because of these base pair rearrangements, the terminal apical loop and the internal bulges which were earlier present in the secondary structure of pre-miR-675 disappeared in the hybrid model (Figure 4A).

However, interactions between H19-exon 2/exon 4 and pre-miR-675 resulted in two different heterodimeric conformations with different binding energies (Appendix A). One model consists of the unperturbed folding dynamics of both H19-exon 2/exon 4 lncRNA and pre-miR-675, thus suggesting that pre-miR-675 does not interact with the exon 2 and exon 4 region of H19 lncRNA (Appendix A). Interestingly, the second model, however, resulted in some localised rearrangements in the base-pairing patterns of both H19 lncRNA and pre-miR-675 (Figure 3B,D and Figure 4B). This demonstrates the occurrence of minute and subtle base pair rearrangements upon the interaction between exon 2/exon 4 of H19 lncRNA and pre-miR-675. However, unlike as mentioned above, the interactions of pre-miR-675 with H19-exon 2 and exon 4 did not resulted in the destabilisation of pre-miR-675 helical conformation (Figure 3B,D). The overall pre-miR-675 was conserved except for the disruption of six GC base-pairing in the basal stem which were previously intact (Figure 4B). Overall, the above study suggests the preference of H19 exon 1/exon 3 for pre-miR-675, thus causing structural rearrangements in the latter. This conformational modulation forms the basis of regulating the activity of pre-miR-675 and modulating its downstream activity through the sequestering activity of H19 lncRNA.

As the controls, two microRNAs miR-1 and miR-122 of independent origin were also used in the co-fold analysis against H19 lncRNA. Both miR-1 and miR-122 fold into a conformation similar to pre-miR-675 (Appendix A). miR-1 is a 71nt miRNA which folds into a stem–loop structure consisting of two loops, namely Loop 1 (6A and 66A) and Loop 2 (23G, 24C and 49A), and one tri-nucleotide bulge (41A, 42G and 43C) before culminating into a pentanucleotide apical loop (Appendix A). miR-122, an 85nt-long miRNA also folds into a stem–loop conformation, which consists of two loops, namely Loop 1 (16G, 17G, 68A and 69A) and Loop 2 (25C and 60A); one bulge (76A); and 12 nucleotide apical loops (Appendix A). The co-folding analysis of miR-1 and miR-122 with H19 lncRNA-exons shows that these miRNAs do not undergo any structural modulations like pre-miR-675 but retain their original conformation (Appendix A). These findings further complement our above results where interaction between H19 lncRNA and pre-miR-675 forces the latter to endure structural modifications which can be the basis of regulatory mechanism to control pre-miR-675 function in the cell.

## 3. Discussion and Conclusions

microRNAs (miRNAs) are known to regulate gene expression and other cellular pathways by influencing the translation and stability of mRNAs. To execute these, miRNAs had to adopt a stable conformation which aids in its interaction with other biomolecules inside the cell. Disruption in microRNA expression/activity results in the development of anomalies like tumours. Determining the structural complexity and interacting partners of any miRNAs will be pivotal in understanding the pathways they are involved in and will further help in determining the progression of cancers. The current study reports the very first secondary structure of pre-miR-675 and its unknown interactors. With RNAcofold and base pair conservation analysis, the mechanism of interaction between H19 lncRNA and pre-miR-675 was also elucidated for the very first time.

Pre-miR-675 folds into a canonical stem loop helical conformation consisting of three internal loops and a bulge which are sandwiched between a basal stem and a terminal apical loop (Figure 1G). The basal stem is highly stabilised due to base pairing interactions between six GC nucleotides. Most of the pre-miR-675 was chemically unmodified, thus suggesting a highly stable conformation (Figure 1G). However, nucleotides comprising loops (especially apical loop nucleotides) and bulges were found to be highly reactive to SHAPE reagent, thus illustrating that they are flexible and single stranded (Figure 1G). Within the cells, these loops/bulges act as the protein-binding sites and thus form the basis of RNA–protein interactions. It is possible that nucleotides present in internal and apical loops of pre-miR-675 act as the site for RNA–protein interactions. Indeed, this study identifies proteins whose binding sites involve nucleotides of Loop II, Loop III and the apical loop of pre-miR-675 (Figure 2G). Further, repeated attempts to understand the folding dynamics of the mature form of pre-miR-675 (miR-675-3p and miR-675-5p) resulted in single stranded conformation. This suggest that both miR-675-3p and miR-675-5p remains flexible within cells as they execute their regulatory effect by binding to the complementary sequences in the 3′UTR of target mRNA.

miRNAs generally interact with the 3′UTR of their target mRNA and confer gene regulation accordingly. However, there are instances where miRNAs also bind to specific RNA-binding proteins (RBP) [23,24]. These interactions are not only useful in the miRNA biogenesis but also have been implicated in the development of several diseases especially tumours, melanomas and neurological disorders [24]. Search with RBPsuite not only resulted in the identification of protein interactors but also determined the protein-binding footprints on pre-miR-675. Amongst all binding sites for FUS, SRSF1, SRSF9, FXR2, LIN28B and HUR proteins were clearly identified on pre-miR-675. Fused in Sarcoma (FUS) protein, an RNA-binding protein is known to aid in gene silencing by simultaneously interacting with both miRNA and mRNA [25]. It has been shown recently that the FUS-induced expression of circular RNA circRHOBTB3 acts as tumour activator and promotes pancreatic ductal adenocarcinoma (PDAC) [26]. FUS and SRSF1 proteins are also responsible for the enrichment of exosomal miRNAs [27,28]. FUS protein identify a specific sequence known as EXOmotif “CGGGAG” in miRNAs, resulting in their transportation to exosomes [28]. Since this study identified FUS as the interactor of pre-miR-675, which also contains the above EXOmotif adjacent to FUS protein-binding motif (Figure 2G), it is quite possible that the binding of FUS to pre-miR-675 could result in the secretion of the latter into exosomes. Indeed, several studies have identified the exosomal presence of pre-miR-675 in many diseases [29,30,31], however, the identification of EXOmotif in pre-miR-675 explains the exosomal properties of pre-miR-675. Lin-28 homolog B (Lin28B) is an evolutionary conserved RNA-binding protein, which can bind to precursor miRNAs and can interfere with their maturation process [32]. The overexpression of Lin28b is linked with the development of ovarian cancer [32] and cellular proliferation in acute myeloid leukaemia [33]. Likewise, HuR is an RNA-binding protein belonging to the embryonic lethal and altered vision (ELAV) family of proteins [34] and is essential for the stability and translation of mRNAs [23]. Interactions between HuR and other miRNAs have been implicated in the development and progression of various cancers [35,36,37]. Furthermore, both Lin28B and HUR had been found to directly interact with H19 lncRNA during gene regulation activities [38,39]. Since pre-miR-675 is derived from H19 lncRNA and shares a similar sequence, it is possible that both Lin28b and HUR RNP can interact with pre-miR-675 during its biogenesis in tumorigenesis. This study, for the first time, reports the binding sites of RNA-binding proteins in pre-miR-675, whereby these interactions are deemed critical for the development of tumorigenesis. Furthermore, the above proteins interact with the nucleotides present in the loops and bulges of pre-miR-675 (Figure 2G). These single-stranded regions within miRNAs have therapeutic importance and serve as an excellent site for designing small molecule inhibitors and Antisense Oligos (ASOs), thus impacting the biological activity of miRNAs. Targeting pre-miR-675 with either small molecule inhibitor/ASOs should be a way to develop novel therapeutic intervention against certain diseases like cancers.

RBPsuite also identified Eukaryotic Initiation Factors (EIF) as a potential interactor for pre-miR-675. Although the exact mechanistic role of pre-miR-675 and EIF is currently unknown, it has been found that miR-141 induced during viral infection targets EIF4E mRNA, resulting in the shutdown of host translation machinery and the promotion of viral proliferation [40]. It is likely that pre-miR-675 expression and activity is regulated during viral infection, leading to cancers—for instance, the hepatitis C virus (HCV) causing hepatocellular carcinoma [41] and the human papilloma virus leading to cervical cancer [42]. However, more studies are required to determine the unique effect of pre-miR-675 during these viral infections, thus making pre-miR-675 a novel anti-viral therapeutic target against above diseases.

Although being expressed from the exon 1 of H19 lncRNA itself, pre-miR-675 base pairs, with different exons of H19 lncRNA, to modulate each other’s activities thus exerting feedback-regulatory mechanisms. In conjunction with RNAcofold and base pair conservation analysis, the current study details co-interaction mechanisms between the individual exons of H19 lncRNA and pre-miR-675. The interactions between pre-miR-675 and exon 1/exon 3 of H19 lncRNA resulted in the opening of pre-miR-675 stem helical structure, as the 5′ end of latter interacts with the 3′ end of former, indicating global rearrangements in pre-miR-675 (Figure 4A,C). Interactions between pre-miR-675 and exon 2/exon 4, however, resulted in localized subtle nucleotide rearrangements in pre-miR-675 (Figure 4B,D). This suggests that pre-miR-675 preferably binds to exon 1 and exon 3 of H19 lncRNA. It is known that the binding of miRNA to lncRNA results in the degradation of lncRNA by mimicking the targets of miRNA [43]. Additionally, the conformational remodelling of pre-miR-675 suggests that H19 lncRNA is also sequestering away pre-miR-675 from its intended target and thus exerting a regulatory effect on pathways involved with pre-miR-675. Indeed, two separate studies have shown that levels of both H19 lncRNA and pre-miR-675 were negatively correlated in lung cancer [44] and human pancreatic ductal carcinoma [45]. Results obtained from co-folding analysis presented in the current study reveals the mechanistic regulatory action of H19 lncRNA on pre-miR-675 by requisitioning it away from its intended targets, thus supporting the feedback regulatory mechanism which H19 lncRNA exerts on pre-miR-675, as observed in the above cases. Further, this study also provides molecular insights into the competitive endogenous RNA (ceRNA) nature of H19 lncRNA, which is well characterized in metabolic disorders [46,47,48], however, its role in cancer development and progression is still elusive. The interaction between H19 lncRNA and pre-miR-675 shows that H19 lncRNA can also act as ceRNA, where it inhibits the gene repression function of pre-miR-675 by interacting with it. Additionally, the cellular system is also enriched with other miRNAs, which can also interact with H19 lncRNA. miR-1 and miR-122 are two such noncoding RNAs, which are originated independently of H19 lncRNA. As a control to our co-folding analysis between H19 lncRNA and pre-miR-675, the co-folding analysis of both miR-1 and miR-122 was performed in the presence of H19 lncRNA (Appendix A). The results obtained show that neither miR-1 nor miR-122 interact with H19 lncRNA while retaining their original conformation. This further strengthens the negative feedback regulatory mechanism/ceRNA nature exerted by H19 lncRNA on pre-miR-675 activity.

Comprehensively, the work described here provides the first detailed analysis of pre-miR-675 stem–loop helical conformation. Based on pre-miR-675 secondary structure, this study allows the identification of previously unknown protein-binding sites, which can be further harnessed to develop novel therapeutics against pre-miR-675. The detailed analysis of the relationship between H19 lncRNA and pre-miR-675 illustrates the preference of pre-miR-675 for exon 1 and exon 3 of H19 lncRNA. Structural modifications in pre-miR-675 are indicative of the sponging effect of H19 lncRNA over pre-miR-675. This contributes to the necessary groundwork for future mutational and genetic analyses and will further aid in the detailed understanding of the mechanism of pre-miR-675 biogenesis, and its function in the development and metastasis of different tumours and cancers.

## 4. Materials and Methods

### 4.1. Pre-miR-675 Conformation Analysis Using Selective 2′ Hydroxyl Acylation Analysed by Primer Extension (SHAPE) Assay

The dsDNA construct of pre-miR-675 was procured as g-blocks from Integrated DNA Technologies (IDT). Each construct is flanked by structural RNA adapters [49] and a T7 promoter region at the 5′ end. These DNA constructs were used as templates for in vitro-transcribed RNA using T7 high yield RNA kit (New England Biolabs, Ipswich, MA, USA). The synthesized RNA was DNase-treated (TURBODNase), purified using Purelink RNA mini kit (Invitrogen, Thermo Fisher Scientific, Waltham, MA, USA) and quantified with nanodrop.

Samples of 6 μg of in vitro-transcribed RNA were denatured at 65 °C for 5 min and snap-cooled in ice. After the addition of a folding buffer (100 mM KCl, 10 mM MgCl_2_, 100 mM Bicine, pH 8.3), RNA was incubated at 37 °C for 10 min. The folded RNA was treated with 10 μL of 5-Nitro Isatoic Anhydride (5NIA, 25 mM final concentration). Subsequently, for negative controls (unmodified RNA), an equivalent amount of dimethyl sulfoxide (DMSO) was added to the folded RNA. The complete reaction mixture was further incubated for 5 min at 37 °C to allow the complete modification of the unpaired RNA nucleotides. Both the modified and unmodified RNAs were purified using the Purelink RNA mini kit and quantified with nanodrop.

Purified RNA from above was reverse-transcribed using a gene-specific reverse primer (Appendix A) directed against the 3′ RNA adapter sequence and SuperScript II reverse transcriptase under error-prone conditions, as previously described [50]. The resultant cDNA was purified using a G50 column (GE healthcare, Chicago, IL, USA) and subjected to second strand synthesis (NEBNext Second Strand Synthesis Module). For library generation, primers specific to the 5′ and 3′ RNA adapter sequence were synthesized (Appendix A) and the whole cDNA was PCR-amplified using the NEB Q5 HotStart polymerase (NEB). Secondary PCR was performed to introduce TrueSeq barcodes [50]. Samples were purified using the Ampure XP (Beckman Coulter, Brea, CA, USA) beads and the resultant libraries were quantified using Qubit dsDNA HS Assay kit (Thermo Fisher Scientific, Waltham, MA, USA). Final libraries were run on an Agilent Bioanalyzer for quality check. These TrueSeq libraries were then sequenced as necessary for their desired length, primarily as paired-end 2 × 151 read multiplex runs on the MiSeq platform (Illumina, San Diego, CA, USA). We used a ShapeMapper2 algorithm [51] to determine the mutation frequency in chemically modified (5NIA) and control (DMSO treated) RNA samples and to calculate chemical reactivity for each RNA nucleotide using the following Equation (1):R = mutr_m_ − mutr_u_(1)
where R is the chemical reactivity, mutr_m_ is the mutation rate calculated for chemically modified RNA and mutr_u_ is the mutation rate calculated for untreated control RNA samples. We used this chemical reactivity to inform a minimum free energy structure using Superfold [52] and visualized the model using VARNA [53]. RNA arc and secondary structure models were generated using RNAvigate [54]. Comparative pre-miR-675 chemical reactivity plots were generated using the plot skyline function from the RNAvigate package [54]. SHAPE reactivities calculated for all the replicates and merged datasets are available in the file *Pre-miR-675_SNRNASM* as a spreadsheet (see data availability statement).

### 4.2. Identification of Protein-Binding Sites on Pre-miR-675

To determine potential interacting partners and to identify the protein-binding sites on the pre-miR-675, searches using the RBPsuite [55] were executed with default parameters. The RBPsuite identifies the protein-binding nucleotides in any given RNA molecule. It uses two-deep learning-based algorithms to identify the RNA-binding protein sites in both linear and circular RNA. The search was performed by selecting pre-miR-675 as linear RNA/circular RNA with the general model being designated as a prediction model. A protein-binding site in an RNA sequence is depicted by a binding score on the scale of 0–1. Higher the score, greater the probability of the presence of RBP site on the queried RNA sequence. RNA motifs with binding scores of less than 0.5 were excluded. If the algorithm identifies a verified motif with binding score of more than 0.5, then those are highlighted in red, and the motif logo is provided in the output.

### 4.3. RNA Co-Fold Analysis

A co-fold analysis between pre-miR-675 and H19 lncRNA exons was performed using RNAcofold algorithm from Vienna RNA websuite [56,57]. RNAcofold calculates hybridization energy and base-pairing probabilities between two interacting RNA molecules. RNAcofold links the two sequences together and the junction is treated as an exterior loop while calculating the hybridization energies. In addition to the minimum free energy structure, the RNAcofold also provides a fraction of suboptimal structures and the equilibrium concentrations of the duplexes formed. Base-pairing conservation was performed to identify the changes in the base-pairing pattern between two interacting RNA molecules.

### 4.4. Statistical Analysis

Most of the statistical analyses were performed using the R software package (version 4.1.1) unless otherwise stated. Pearson’s correlation test was used to evaluate the correlation between chemically probed replicates (*n* = 2) across all test samples.

## Figures and Tables

**Figure 1 ncrna-09-00045-f001:**
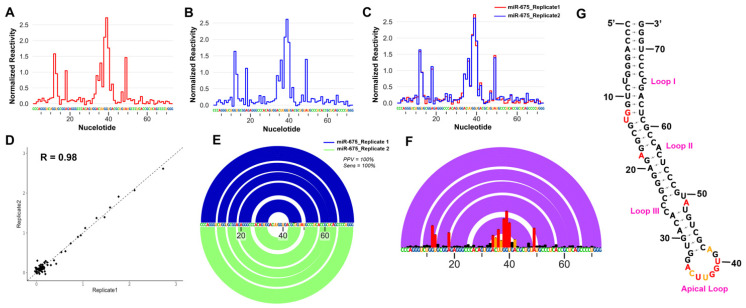
In vitro architecture and statistical analysis of pre-miR-675 secondary structure. (**A**) Normalized 5-NIA reactivity profile for in vitro synthesized pre-miR-675 replicate 1. (**B**) Normalized 5-NIA reactivity profile for in vitro synthesized pre-miR-675 replicate 2. Some of the nucleotides were found to have high preference for 5-NIA as compared to others in both the replicates. (**C**) Overlapped normalized 5-NIA reactivity profile of in vitro synthesized pre-miR-675 replicates. The pre-miR-675-replicate 1 is represented in red, while pre-miR-675-Replicate 2 is represented as blue. The reactivity profile was found to be highly similar for both the replicates. (**D**) Scatter plot with Pearson’s correlation coefficient between the normalized reactivity profiles of pre-miR-675 replicates, R = 0.98 illustrates identical and reproducible reactivity profiles between both the replicates of pre-miR-675. (**E**) Overlapped RNA arcplot (depicting secondary structure) with Scorer results for pre-miR-675, Replicate 1 and Replicate 2. Both pre-miR-675 replicates fold into identical conformation with PPV = 100% and SENS = 100%. (**F**) Normalized 5-NIA reactivity profile generated after averaging the reactivity profiles of pre-miR-675 replicate 1 and replicate 2. Also shown is the overlapped arc profile representing the secondary structure of pre-miR-675. (**G**) Secondary structure model of pre-miR-675 was obtained after feeding superfold with obtained SHAPE constraints. Black colour represents 5-NIA reactivity < 0.4 which means nucleotides are base paired, orange represents 5-NIA reactivity between 0.4 and 0.8 meaning these nucleotides are either unpaired or base paired and red represents 5-NIA reactivity > 0.8 for nucleotides which means these nucleotides are unpaired and not involved in base pairing.

**Figure 2 ncrna-09-00045-f002:**
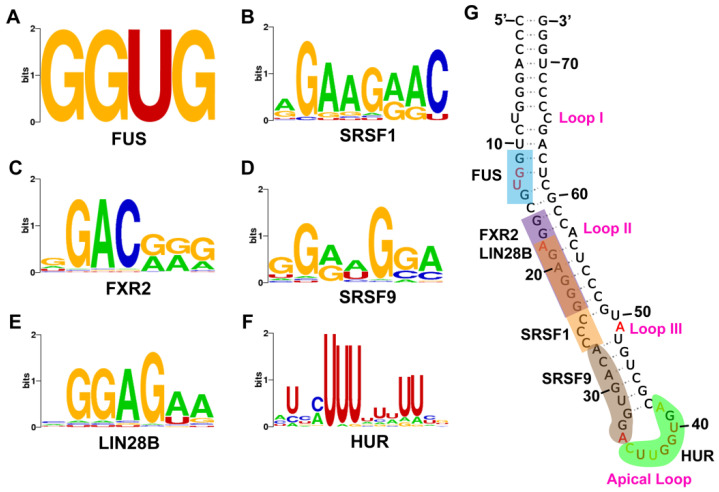
Pre-miR-675 protein-binding sites of different RNA-binding proteins identified through RBPsuite. Pre-miR-675 binding sequences identified for proteins. (**A**) Fused in Sarcoma (FUS). (**B**) Serine/arginine-rich splicing factor 1 (SRSF1). (**C**) FMR1 autosomal Homolog 2 (FXR2). (**D**) Serine/arginine-rich splicing factor 9 (SRSF9). (**E**) Lin-28 homolog B (Lin28B). (**F**) Human Antigen R (HuR). For better representation, nucleotides in the sequence logo is represented by different colours with Adenine as green, Guanine as orange, Uracil as red and Cytosine as blue. (**G**) Protein-binding sites in pre-miR-675. Most of the protein-binding sites in pre-miR-675 include the nucleotides from the loop region thus making pre-miR-675 an excellent target of therapeutic importance.

**Figure 3 ncrna-09-00045-f003:**
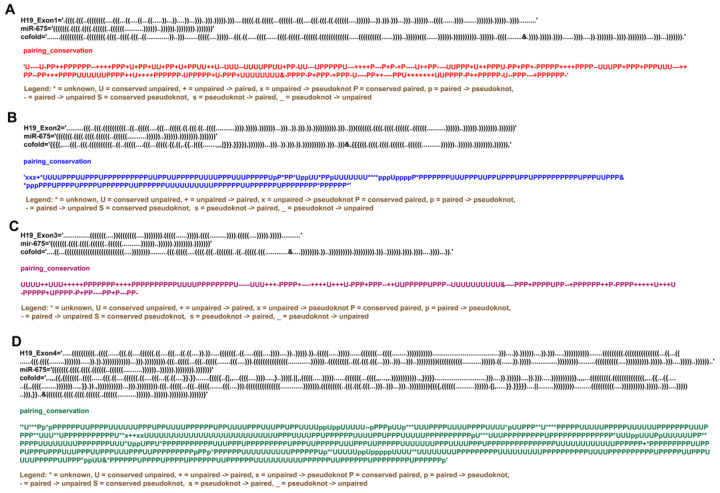
RNA co-fold and base pair conservation analysis between pre-miR-675 and H19-exons. (**A**) Exon 1 (Model 1). (**B**) Exon 2 (Model 2). (**C**) Exon 3 (Model 1). (**D**) Exon 4 (Model 2) of precursor H19 lncRNA. Upper panel in each figure represents the folding dynamics of individual H19-exons, pre-miR-675 and their co-folding pattern analysed through RNAcofold in dot–bracket notation. The “&” in the co-fold pattern of the hybrid structure is the separator between two RNA molecules. The middle panel represents the H19-exons and pre-miR-675 base pairs, which were either modified or remained unchanged during the co-folding of both the RNA molecules. The lower panel explains the meaning of each symbol/notations used to determine the base pair conservation during the co-folding process of H19-exons and pre-miR-675. pre-miR-675 has been shown to prefer exon 1 and exon 3 of H19 lncRNA, resulting in the opening of its canonical stem–loop helical structure, while its interaction with exon 2 and exon 4 results more in localized rearrangements in base-pairing between nucleotides.

**Figure 4 ncrna-09-00045-f004:**
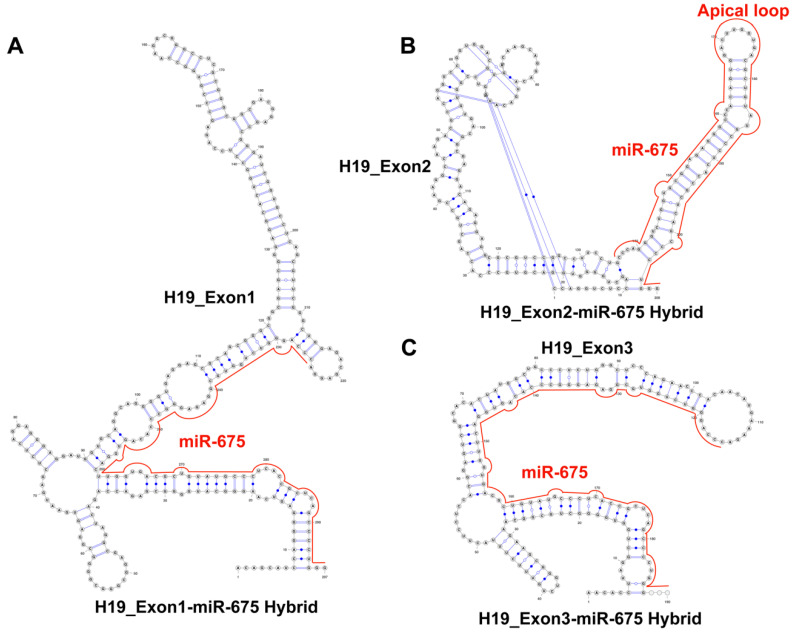
Structural modifications in pre-miR-675 enables it to bind H19-exon1/exon3. Hybrid RNA conformer between (**A**) H19-exon 1 and pre-miR-675, (**B**) H19-exon 2 and pre-miR-675 and (**C**) H19-exon 3 and pre-miR-675. The hybrid conformation between H19-exon 1 and exon 3 with pre-miR-675 (red line) is formed due to the disruption of pre-miR-675 stem–loop helical conformation and formation of new base pairs between two RNA molecules, while the pre-miR-675 (red line) retains its canonical conformation, when interacting with exon 2 of H19 lncRNA, thus retaining all its bulges and loops, including the apical loop. These hybrid conformations explain the negative feedback regulatory mechanism exerted by H19 lncRNA and pre-miR-675 on each other during the development and progression of cancer.

## Data Availability

Pre-miR-675 shape reactivities are available as *Pre-miR-675_SNRNASM* through following link: https://docs.google.com/spreadsheets/d/1HEIw_xVfhZcXcYlLUSar7ZlenY7S5XKt/edit?usp=share_link&ouid=116170607567213620520&rtpof=true&sd=true (accessed on 7 August 2023).

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
