# Peer review of "Structural Modifications and Novel Protein-Binding Sites in Pre-miR-675—Explaining Its Regulatory Mechanism in Carcinogenesis"

_ncrna, 2023, doi:10.3390/ncrna9040045_

Round 1
Reviewer 1 Report
Significance:
MS by Abhishek Dey aimed to study the folding dynamics of miR- 675 and identify its interacting proteins and crosstalk with its parent lncRNA H19. The question they are trying to answer is interesting and relevant in the ncRNA field but here are some points that the author can highlight in the MS.
Comments:
· H19 is a precursor lncRNA for miR-675 therefore it is wise to check the SHAPE RNA structure prediction for H19.
· Author tried to identify the structural modifications in miR-675 which enables it to bind H19-exons. As a control, it will be nice to see how miRNAs of independent origin interact with H19 lncRNA which mimics the cell milieu in an absolute way.
· Authors can comment on how this study perceives the ceRNA nature of H19 in miR-675 regulation.
· Author discusses Figure 1H in the results and discussion but Figure 1 only has a panel till 1G. might be mislabelling in Figure 1. I would like the author to correct it to avoid confusion.
Author Response
- H19 is a precursor lncRNA for miR-675 therefore it is wise to check the SHAPE RNA structure prediction for H19.
- Author Response:
We completely agree with the reviewer about performing SHAPE assay for H19 lncRNA, however due to funding constraints we were unable to perform this experiment for this study. Plans are already in place to perform SHAPE analysis along with other biochemical analysis on H19 lncRNA for future analysis. Nevertheless, being derived from H19 lncRNA, our study reports the absolute conformation of miR-675 along with the structural modulations it undergoes while interacting with H19 lncRNA. This study provides an insight into the regulatory mechanism executed by both H19 and miR-675 for each other’s activity during cancer progression and development.
- Author tried to identify the structural modifications in miR-675 which enables it to bind H19-exons. As a control, it will be nice to see how miRNAs of independent origin interact with H19 lncRNA which mimics the cell milieu in an absolute way.
- Author Response:
We are thankful to the reviewer for this suggestion. A similar RNA co-fold interaction study between H19 exons and miR-1 and miR-122 (Supplementary Figure S3 and S4) as controls has now been performed and described in the main text (Section 2.3: Line 278-289 and Discussion: Line 393-396). Both miR-1 and miR-122 have origin independent of H19 lncRNA. Co-folding analysis suggest that both miR-1 and miR-122 does not undergo any structural rearrangements like pre-miR-675 while interacting with any of the exons of H19 lncRNA and retain their original conformation. This analysis further complements our finding about the restructuring of pre-miR-675 conformation as a process of negative feedback regulation exerted by H19 lncRNA on pre-miR-675.
- Authors can comment on how this study perceives the ceRNA nature of H19 in miR-675 regulation.
- Author Response:
We are grateful to the reviewer for this suggestion. Competitive endogenous RNA (ceRNA) is lncRNA which can regulate gene expression by acting as molecular sponges by targeting miRNAs and sequestering them away from their intended mRNA targets. This mechanism inhibits the binding of miRNA to target mRNA and their decay thus ensuring gene expression. We have already described this nature of H19 lncRNA in our manuscript in subtle manner (Line 378-380 and Line 384-385) in discussion section, however now we provided more explanation on the sponging activity of H19 lncRNA with suitable references in discussion section (Line 386-392). ceRNA nature of H19 lncRNA has been extensively studied in various metabolic disorders like cardiovascular diseases, hyperglycemia, and endocrine disorder. However, its exact mechanism and role in cancer development and progression is yet to be elucidated. This study explains the mechanism by which the sponging activity (ceRNA nature) of H19 lncRNA can bind to miR-675 and restrict the later for its regulatory pathways.
- Author discusses Figure 1H in the results and discussion but Figure 1 only has a panel till 1G. might be mislabelling in Figure 1. I would like the author to correct it to avoid confusion.
- Author Response:
We have now rectified the mistake in the entire manuscript and have replaced Figure 1H with Figure 1G.

Reviewer 2 Report
The manuscript by Abhishek Dey entitled “Structural modifications and novel protein binding sites in miR-675 explaining its regulatory mechanism in carcinogenesis” is scientifically convincing and well-conducted. However, the introduction has inadequate information about the previous study findings for readers to follow the present study rationale. Further, I have these concerns that readers may have
comments
1. Author has revealed that binding sites for FUS, SRSF1, SRSF9, FXR2, LIN28B, and 273 HUR proteins were identified on miR-675. Is there any available literature or did you perform any in vitro assays in support of your Insilco results?
2. A single microRNA has many putative targets. Is there any possibility that miR-675 binding proteins like FUS, SRSF1, etc. play a role in the specificity of a particular mRNA target at 3’UTR region?
3. Authors should disclose whether miR-675 binding proteins or the H19 long noncoding RNA (lncRNA) play any role in degerming the nature of miR-675 (endosomal or exosomal miRNA)
Needs minor editing
Author Response
However, the introduction has inadequate information about the previous study findings for readers to follow the present study rationale.
Author Response
- We have now expanded the introduction by providing additional information regarding the protein binding nature of miR-675 and its crosstalk with H19 lncRNA (Introduction section: Line 72-86) with appropriate references. We have also illustrated the need to understand the unknown molecular mechanism that are driving these interactions which should now provide rationale to the current study.
Further, I have these concerns that readers may have
Comments
- Author has revealed that binding sites for FUS, SRSF1, SRSF9, FXR2, LIN28B, and 273 HUR proteins were identified on miR-675. Is there any available literature or did you perform any in vitroassays in support of your Insilco results?
- Author Response:
- We thank reviewer for this question. Search for Protein binding motif and Ribonuceloprotein (RNP) partners for miR-675 was executed using RBPsuite which uses two deep-learning algorithms to identify the protein binding site on any RNA (discussed in materials and methods section of manuscript). Majority of the protein identified are well characterized RNP with known RNA binding motifs. Since these regions were also recognized through RBPsuite it can be concluded that miR-675 does harbours the binding site for these different proteins. Our repeated searches failed to retrieve any direct evidence of interaction between miR-675 and the proteins reported here. However, we did find two independent studies where the RNP HUR (Keniry, A.et al. The H19 lincRNA is a developmental reservoir of miR-675 that suppresses growth and Igf1r. Cell Biol.2012, 14, 659–665), and Lin28B (Helsmoortel HH,et al. LIN28B is over-expressed in specific subtypes of pediatric leukemia and regulates lncRNA H19. Haematologica. 2016 Jun;101(6):e240-4) interact directly with H19 lncRNA. Since miR-675 is derived from H19 lncRNA and have similar sequences it is possible that miR-675 can also interact with HUR and Lin28B. We have now added these observations in discussion section (Discussion: Line 343-347).
- A single microRNA has many putative targets. Is there any possibility that miR-675 binding proteins like FUS, SRSF1, etc. play a role in the specificity of a particular mRNA target at 3’UTR region?
- Author Response:
- In general, microRNAs consist of nucleotide sequences complimentary to the 3’ UTR regions of target mRNA through which they interact and regulates gene expression. Binding of RNP on miRNA has known to have multiple effects (Ciafrè SA, Galardi S. microRNAs and RNA-binding proteins: a complex network of interactions and reciprocal regulations in cancer. RNA Biol. 2013;10:935-42). It can compete with miRNA for the target mRNA 3’UTR binding site thus shielding the mRNA from degradation, can interact with miRNA and enhance its activity by helping in repressing the mRNA expression, and involved in miRNA biogenesis. HUR is one of the RNP which has known to have all the three above effects on miRNA. Binding of HUR to H19 lncRNA liberates miR-675 (Keniry, A.; Oxley, D.; Monnier, P.; Kyba, M.; Dandolo, L.; Smits, G.; Reik, W. The H19 lincRNA is a developmental reservoir of miR-675 that suppresses growth and Igf1r. Cell Biol.2012, 14, 659–665), while interaction between HUR and let-7 helps let-7 to bind 3’ UTR of c-Myc and repress the later expression (Ciafrè SA, Galardi S. microRNAs and RNA-binding proteins: a complex network of interactions and reciprocal regulations in cancer. RNA Biol. 2013;10:935-42). Though we were unable to identify the consequences of interactions between miR-675 binding RNPs on 3’UTR of specific mRNA, nonetheless, based on the above study of HUR RNP, it is possible that miR-675 interacting RNPs are either involved in enhancing the function of miR-675 in gene repression or more involved in the biogenesis of miR-675. However, more biochemical and in cell studies will be required to determine the nature and extent of these interactions.
- Authors should disclose whether miR-675 binding proteins or the H19 long noncoding RNA (lncRNA) play any role in degerming the nature of miR-675 (endosomal or exosomal miRNA)
- Author Response:
- We again thank reviewer for pointing our attention towards the exosomal nature miR-675. Though we didn’t come across any study assessing the secretion of miR-675 in extracellular vesicles due to the bound proteins, however, we did identify one study describing the presence of miR-675 in the exosomes during certain disease state like osteosarcoma (Gong L., Bao Q., Hu C., Wang J., Zhou Q., Wei L., et al. (2018). Exosomal miR-675 from Metastatic Osteosarcoma Promotes Cell Migration and Invasion by Targeting CALN1. Biochem. Biophys. Res. Commun. 500 (2), 170–176). In independent studies, it was also found that both SRSF1 and FUS protein aid in the enrichment of exosomal miRNA (Xu YF, Xu X, Gin A, Nshimiyimana JD, Mooers BHM, Caputi M, Hannafon BN, Ding WQ. SRSF1 regulates exosome microRNA enrichment in human cancer cells. Cell Commun Signal. 2020 Aug 20;18(1):130, Garcia-Martin R, Wang G, Brandão BB, Zanotto TM, Shah S, Kumar Patel S, Schilling B, Kahn CR. MicroRNA sequence codes for small extracellular vesicle release and cellular retention. Nature. 2022 Jan;601(7893):446-451). FUS protein identify EXOmotif “CGGGAG” in miRNA and help in the transportation of miRNA to exosomes. Since, FUS protein has been identified as an interactor for miR-675 which also contains this EXOmotif “CGGGAG” (right next to the FUS binding site in miR-675, Figure 2G), it is possible that binding of FUS or other RNPs can enhance the transportation of miR-675 to exosomes during carcinogenesis. However, more biochemical analysis is required to ascertain the same. Nevertheless, the above findings can make miR-675 an excellent target to use as biomarker in the diagnosis of cancer. We have now included this observation in discussion section (Discussion: Line 328-336).
Comments on the Quality of English Language
Needs minor editing
Author Response
- We extensively and repeatedly reviewed the manuscript and rectified any errors or mistakes if present in manuscript.

Reviewer 3 Report
The paper is interesting, however, several arrangements are needed before its publication.
In line 27-29, where it reads “miR-675” should be changed by “Pre-miR-675”. In general revise all the manuscript and correct mature miRNA form to Precursor miRNA, if it has been processed by Dicer it should be named miR-675 but if it hasn’t it most named Pre-miR-675.
In line 39 delete reference 7 because you use this cite in line 41 and no other reference it is cited.
In line 55 delete reference 11 because you use this cite in line 57 and no other reference it is cited.
In line 57-59 Please check the sentence “Interaction of mir-675-5p with serine-threonine phosphatases have been shown to activate Glycogen Synthase Kinase 3êžµ (GSK 3êžµ), thus enhancing the nuclear localization of êžµ-catenin which aids in the progression of colorectal cancer [12].” Serine-threonine phosphatases should be changed from position or erased.
In line 59-61 You mention “Direct interaction between miR-675 and mRNA of ubiquitin 3 ligase family (c-Cbl and Cbl-b) have shown to enhance tumorigenesis and metastasis in breast cancer cells [13]”, however, this effect is achieved because downregulation of c-Cb1 and Cbl-b is reached not because of the interaction per se. Can you argue about it please.
In line 75-76 You mention “Here in this study the very first detailed conformational analysis of miR-675 by Selective 2’ Hydroxyl Acylation analysed by Primer Extension (SHAPE) assay is reported.” I suggest to delete “assay is reported”.
Also, the prepositions “Here” and “in this” are redundant in the same sentence.
5NIA should be defined the first time of appearance.
Figure 1H is not in the manuscript but it is mentioned all over the work.
In line 208, the word “re-arrangements” should not be written without middle dash, please, correct it.
In line 297-300 the citations 28 and 29 in the text do not correspond to the information provided. “It is likely that miR-675 expression and activity is regulated during viral infection leading to cancers like Hepatitis C virus (HCV) causing hepatocellular carcinoma [28] and human papilloma virus leading to cervical cancer [29], miR-675 is not mentioned in these papers.
In line 309-311 You mention Figure 5 but it is not found in the manuscript, please add the figure or correct the text.
In line 316-319 The citations do not correspond to those used in the manuscript. “Indeed, two separate studies have shown that levels of both H19 lncRNA and miR-675 were negatively correlated in lung cancer (Matouk I, The non-coding RNAs, 2015) and human pancreatic ductal carcinoma (Ma L, h19 derived, 2018).”
Author Response
In line 27-29, where it reads “miR-675” should be changed by “Pre-miR-675”. In general revise all the manuscript and correct mature miRNA form to Precursor miRNA, if it has been processed by Dicer it should be named miR-675 but if it hasn’t it most named Pre-miR-675.
- Author Response: We thank reviewer for this suggestion and have made necessary corrections in the entire manuscript. miR-675 has now been replaced by pre-miR-675 in the entire manuscript.
In line 39 delete reference 7 because you use this cite in line 41 and no other reference it is cited.
- Author Response: As per suggestion we have deleted reference 7 from line 39 in the manuscript.
In line 55 delete reference 11 because you use this cite in line 57 and no other reference it is cited.
- Author Response: As per suggestion reference 11 has been deleted from line 55.
In line 57-59 Please check the sentence “Interaction of mir-675-5p with serine-threonine phosphatases have been shown to activate Glycogen Synthase Kinase 3êžµ (GSK 3êžµ), thus enhancing the nuclear localization of êžµ-catenin which aids in the progression of colorectal cancer [12].” Serine-threonine phosphatases should be changed from position or erased.
- Author Response: Study conducted by Saieva et al (L. Saieva et al., Hypoxia-Induced miR-675-5p Supports β-Catenin Nuclear Localization by Regulating GSK3-β Activity in Colorectal Cancer Cell Lines. Int J Mol Sci 21, (2020)) have reported PPP2CA, and PPP2R2B, a serine/threonine phosphatase as a probable target of miR-675-5p while regulating the expression of Glycogen Synthase Kinase 3êžµ in the progression of colorectal cancer. Real time PCR experiments have shown that in the presence of transfected miR-675-5p, the mRNA levels of PPP2CA, and PPP2R2B reduces resulting in the GSK3β activation by Ser9 residue de-phosphorylation. This study proves the indirect involvement of miR-675-5p in regulating GSK3β through serine-threonine phosphatases and hence being appropriately cited in the manuscript.
In line 59-61 You mention “Direct interaction between miR-675 and mRNA of ubiquitin 3 ligase family (c-Cbl and Cbl-b) have shown to enhance tumorigenesis and metastasis in breast cancer cells [13]”, however, this effect is achieved because downregulation of c-Cb1 and Cbl-b is reached not because of the interaction per se. Can you argue about it please.
- Author Response: In the above-mentioned study (C. Vennin et al., H19 non coding RNA-derived miR-675 enhances tumorigenesis and metastasis of breast cancer cells by downregulating c-Cbl and Cbl-b. Oncotarget 6, 29209-29223 (2015)),investigator have performed the sequence alignment analysis between miR-675 and c-Cb1 and Cbl-b mRNAs and found that miR-675 has a complementary sequence present in the mRNA of c-Cb1 and Cbl-b (Figure 1a, Venin et al) resulting in an alignment of c-Cb1 and Cbl-b to miR-675. Furthermore, they performed biochemical experiments to correlate the expression level of c-Cb1 and Cbl-b in breast cancer cell by overexpressing H19 lncRNA. The later analysis reveals the negative correlation between the expression levels of c-Cb1 and Cbl-b and H19 RNA levels (Figure 1, Venin et al). This suggest that H19 derived miR-675 can interact directly with the c-Cb1 and Cbl-b mRNA through complementary sequence thus downregulating their protein expression and in turn aiding in metastasis process in breast cancer cells.
In line 75-76 You mention “Here in this study the very first detailed conformational analysis of miR-675 by Selective 2’ Hydroxyl Acylation analysed by Primer Extension (SHAPE) assay is reported.” I suggest to delete “assay is reported”.
- Author Response: As per suggestion of the reviewer we have now modified the above statement in the manuscript. It now reads like this “Here we report the very first detailed conformational analysis of pre-miR-675 using Selective 2’ Hydroxyl Acylation analysed by Primer Extension (SHAPE) assay.” (Line 98-100)
Also, the prepositions “Here” and “in this” are redundant in the same sentence.
- Author Response: As per the suggestion of the reviewer “In this” has now been deleted from line 75 of manuscript. (Line 98)
5NIA should be defined the first time of appearance.
- Author Response: Full name of 5NIA has now been described in the manuscript. (Line 113)
Figure 1H is not in the manuscript but it is mentioned all over the work.
- Author Response: We have now rectified the mistake in the entire manuscript and have replaced Figure 1H with Figure 1G.
In line 208, the word “re-arrangements” should not be written without middle dash, please, correct it.
- Author Response: Necessary correction has now been made in the main text and middle dash is removed from the word re-arrangements. (Line 243)
In line 297-300 the citations 28 and 29 in the text do not correspond to the information provided. “It is likely that miR-675 expression and activity is regulated during viral infection leading to cancers like Hepatitis C virus (HCV) causing hepatocellular carcinoma [28] and human papilloma virus leading to cervical cancer [29], miR-675 is not mentioned in these papers.
- Author Response: We are grateful to the reviewer for this comment and now have replaced these citations with the new and more relevant references in the manuscript. (Line 261-362)
In line 309-311 You mention Figure 5 but it is not found in the manuscript, please add the figure or correct the text.
- Author Response: We have now performed necessary changes in the entire manuscript with Figure 5 has now been reannotated as Figure 4.
In line 316-319 The citations do not correspond to those used in the manuscript. “Indeed, two separate studies have shown that levels of both H19 lncRNA and miR-675 were negatively correlated in lung cancer (Matouk I, The non-coding RNAs, 2015) and human pancreatic ductal carcinoma (Ma L, h19 derived, 2018).”
- Author Response: The above citations have now been added to the reference section of the manuscript. (Line 382-383)

Round 2
Reviewer 3 Report
In line 72-73 You used the term abiotic “Similarly, repression of several miRNA biogenesis component due to abiotic stress causes the onset of hepatocellular carcinoma (16).” This term is not mentioned in the reference 16 and is normally used in plants not in human, please correct.
In your study you are using the pre-miR form ( Ì´70nt), however, it is not the case for all the references used in your manuscript, were they used miRNA form ( Ì´22nt). Please, check and correct basing on the form used in the references and in your results respectively.
Author Response
Authors response:
The author thanks reviewer for reviewing the manuscript and providing with valuable suggestion to improve it. We have now uploaded the revised version of the manuscript. We have tried to incorporate the suggestion provided by the reviewer and provided point by point responses for the concerns/comments raised by the reviewer. Author responses are in italics.
Comments and Suggestions for Authors
1) In line 72-73 You used the term abiotic “Similarly, repression of several miRNA biogenesis component due to abiotic stress causes the onset of hepatocellular carcinoma (16).” This term is not mentioned in the reference 16 and is normally used in plants not in human, please correct.
- Generally abiotic stress is related to any non-living components like UV radiation, heavy metals etc causing changes in the living organism (plants, animals etc). The aforementioned work (N. Kitagawa et al., Downregulation of the microRNA biogenesis components and its association with poor prognosis in hepatocellular carcinoma. Cancer Sci 104, 543-551 (2013)) describes that the combination of alcohol and smoking can causes dysregulation in miRNA biogenesis. Since both alcohol and smoking causes stress conditions hence these were collectively mentioned as abiotic stress. However for better clarity the term “abiotic stress” has been replaced with more specific terms “alcohol and smoking” as they have been mentioned in the cited reference. (Line 73)
2) In your study you are using the pre-miR form ( Ì´ 70nt), however, it is not the case for all the references used in your manuscript, were they used miRNA form ( Ì´ 22nt). Please, check and correct basing on the form used in the references and in your results respectively.
- miR-675-5p and miR-675-3p are ~21 nt matured forms which are originated from the 73 nt long pre-miR-675 from its different arms after processing through DICER. Though these are the functional forms of pre-miR-675, their activity can be controlled by regulating pre-miR-675 itself. In the current study conformational analysis was simultaneously performed on the pre-miR-675 and its mature forms. pre-miR-675 folded into a stem-loop helical conformation however we found that the mature forms remain single stranded. Thus we concluded that both miR-675-5p and miR-675-3p does not have any specific architecture and execute their function by binding to the 3’UTR of their target RNA (Discussion section: Line 299-302). This study also provides an insight into the regulatory mechanism executed by both H19 and miR-675 on each other’s activity during cancer progression and development via the regulation of pre-miR-675 biogenesis into its mature forms miR-675-5p and miR-675-3p. Indeed a recent study have shown the negative effect of miR-675 on DUX4 expression through the processing of former into its mature form in muscular dystrophy (Saad, N.Y., Al-Kharsan, M., Garwick-Coppens, S.E. et al. Human miRNA miR-675 inhibits DUX4 expression and may be exploited as a potential treatment for Facioscapulohumeral muscular dystrophy. Nat Commun 12, 7128 (2021). https://doi.org/10.1038/s41467-021-27430-1). Based on the above results and observation, I believe that the term for pre-miR-675 has been appropriately used in the manuscript.
